# Spatially distributed infection increases viral load in a computational model of SARS-CoV-2 lung infection

Melanie E. Moses[1,2]*, Steven Hofmeyr[3], Judy L. Cannon[4], Akil Andrews[1], Rebekah Gridley[4], Monica Hinga[1], Kirtus Leyba[5], Abigail Pribisova[1], Vanessa Surjadidjaja[1], Humayra Tasnim[1], Stephanie Forrest[2,5]

**1** Department of Computer Science, University of New Mexico, Albuquerque, New Mexico, United States of America, **2** Santa Fe Institute, Santa Fe, New Mexico, United States of America, **3** Lawrence Berkeley National Laboratory, Berkeley, California, United States of America, **4** Department of Molecular Genetics and Microbiology, University of New Mexico School of Medicine, Albuquerque, New Mexico, United States of America, **5** Biodesign Institute, Arizona State University, Tempe, Arizona, United States of America

* melaniem@unm.edu

**Data Availability Statement:** All code can be accessed at https://github.com/ AdaptiveComputationLab/simcov. Configuration

## Abstract

A key question in SARS-CoV-2 infection is why viral loads and patient outcomes vary dramatically across individuals. Because spatial-temporal dynamics of viral spread and immune response are challenging to study in vivo, we developed Spatial Immune Model of Coronavirus (SIMCoV), a scalable computational model that simulates hundreds of millions of lung cells, including respiratory epithelial cells and T cells. SIMCoV replicates viral growth dynamics observed in patients and shows how spatially dispersed infections can lead to increased viral loads. The model also shows how the timing and strength of the T cell response can affect viral persistence, oscillations, and control. By incorporating spatial interactions, SIMCoV provides a parsimonious explanation for the dramatically different viral load trajectories among patients by varying only the number of initial sites of infection and the magnitude and timing of the T cell immune response. When the branching airway structure of the lung is explicitly represented, we find that virus spreads faster than in a 2D layer of epithelial cells, but much more slowly than in an undifferentiated 3D grid or in a well-mixed differential equation model. These results illustrate how realistic, spatially explicit computational models can improve understanding of within-host dynamics of SARS-CoV-2 infection.

## Author summary

A key question in SARS-CoV-2 infection is why viral loads and patient outcomes are so different across individuals. Because it's difficult to see how the virus spreads in the lungs of infected people, we developed Spatial Immune Model of Coronavirus (SIMCoV), a computational model that simulates hundreds of millions of cells, including lung cells and immune cells. SIMCoV simulates how virus grows and then declines, and the simulations match data observed in patients. SIMCoV shows that when there are more initial infection sites, the virus grows to a higher peak. The model also shows how the timing of the

files to reproduce each figure in the paper are available in the configs folder in that repository.

**Funding:** This project was funded by NSF (www. nsf.gov) 2030037 and 2029696 which includes Coronavirus Aid, Relief, and Economic Security (CARES) Act funding (MEM, JLC, AA, RG, MH, KL, AP, VS, HT, SF). DARPA (www.darpa.mil) provided partial funding through AFRL FA-8650-18-C-6898 (JLC, MM, HT and AA). JLC is also supported by the Autophagy Inflammation and Metabolism Center of Biomedical Research Excellence (AIM CoBRE, NIH NIGMS P20GM121176). This work is also partly supported by the Advanced Scientific Computing Research (ASCR) program within the Office of Science of the DOE (www.energy.gov) under contract number DE-AC02-05CH11231 (SH) and the Exascale Computing Project(17-SC-20-SC), a collaborative effort of the U.S. Department of Energy Office of Science (SH). The funders had no role in study design, data collection and analysis, decision to publish, or preparation of the manuscript.

**Competing interests:** The authors have declared that no competing interests exist.

immune response, particularly the T cell response, can affect how long the virus persists and whether it is ultimately cleared from the lungs. SIMCoV shows that the different viral loads in different patients can be explained by how many different places the virus is initially seeded inside their lungs. We explicitly add the branching airway structure of the lung into the model and show that virus spreads slightly faster than it would in a 2D layer of lung cells, but much slower than in traditional mathematical models based on differential equations. These results illustrate how realistic spatial computational models can improve understanding of how SARS-CoV-2 infection spreads in the lung.

## Introduction

Reducing the spread, severity, and mortality caused by severe acute respiratory syndrome coronavirus 2 (SARS-CoV-2) infection is an urgent global priority. A key question is why viral loads and patient outcomes vary so dramatically among infected individuals. While severe COVID-19 disease is correlated with some known risk factors, such as age, co-morbidities, and some immune response characteristics [1–3], it remains challenging to understand why some patients develop high viral loads and others do not.

We developed a scalable computational model to study the role of spatial effects in determining the time course of viral load within patients, which has been shown to affect disease severity [4] and transmission [5]. Most prior work uses Ordinary Differential Equation (ODE) models to represent within-host virus dynamics, e.g., [6–12]. Such models are useful for studying the onset and duration of the infective period [13] and the effect of various therapeutics given at different times [14–16]. However, they have limited ability to fully account for dynamics in the large and complex structure of the lung [12, 17]. The Spatial Immune Model of Coronavirus (SIMCoV) is an agent-based model (ABM) that simulates infection dynamics and CD8+ T cell responses in tissue consisting of hundreds of millions of epithelial cells. Our model results highlight the importance of spatial viral dispersion in the lung and how this affects peak viral load. We focus on CD8+ T cells because they are a key player in the immune response to SARS-CoV-2 and because spatial dynamics are important for these cells that move in the lung to kill virally infected cells. Earlier studies report that T cell levels are correlated with disease severity [18], and there is mounting evidence that T cells are critical for protection from severe illness and for preventing subsequent infection [19–22]. These studies collectively suggest that an early and strong T cell response is correlated with less severe disease, but they do not provide a quantitative understanding of how T cell response impacts viral dynamics.

SIMCoV and other ABMs are appealing because they can represent and visualize the complex heterogeneous movements of individual cells and diffusing fields of small particles (i.e., virions or cytokines) [23]. Our prior work analyzed how key factors such as spread of infection, signals, and cells affect search efficiency, timing of immune response, and clearance of infection; this work showed that interactions that depend on movement and physical cell-cell contact require a spatially explicit modeling framework [24, 25]. The SIMCoV simulation of SARS-CoV-2 reinforces the importance of spatial effects as drivers of viral dynamics in the lung.

Our simulations replicate the viral dynamics reported from infected patients and explain differences among those patients in terms of the initial number of foci of infection (FOI) and the magnitude and timing of the CD8+ T cell response. SIMCoV demonstrates how the spatial structure of the lung can affect viral spread in the tissue and contribute to the wide diversity of outcomes and infection dynamics observed in COVID-19 patients.

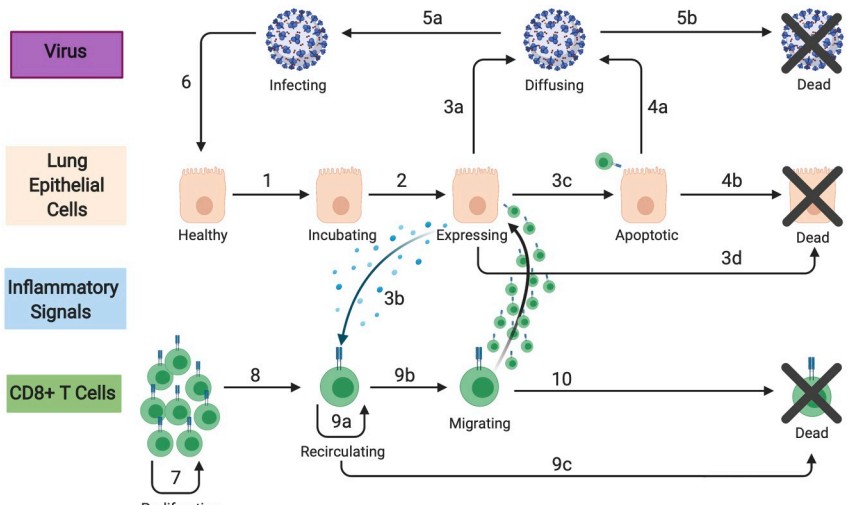

**Fig 1. SIMCoV model components and their interactions.** Epithelial and T cells are represented as agents; virions and inflammatory signals are represented as concentrations. Numbered transitions are described in Materials and Methods.

## Results

### SIMCoV model overview

We initialize SIMCoV with SARS-CoV-2 virus that can infect lung epithelial cells. In most of our experiments, epithelial cells are modeled as a 2D grid, where each grid point represents a 5 x 5 x 5 $\mu$m$^3$ volume. Upon infection, epithelial cells incubate virus in what is often called an eclipse phase, and then they express virus until they die. Expressing cells produce both virus and an inflammatory signal, which is an abstraction of the cytokines that influence T cell extravasation into tissue. The virus and inflammatory signal both diffuse through tissue. After a delay, an abstract pool is created that represents CD8$^+$ cytotoxic T cells that are activated in lymph nodes and then circulate in the vasculature. When T cells reach the lung and encounter a concentration of inflammatory signal above a threshold, they extravasate and are then explicitly represented as mobile agents in SIMCoV. We adjust the probability of T cell extravasation according to the fraction of the lung that is modeled in the simulation. T cells that have extravasated move at random across the 2D grid until they either die or encounter an epithelial cell incubating or expressing virus, in which case the T cells cause apoptosis of the infected epithelial cell with some probability. SIMCoV extends an ABM developed for influenza [25–27] by adding the ability to simulate viral spread through the 3D branching structure of lung epithelial cells and parameterizing the model for SARS-CoV-2. Each transition in Fig 1 is parameterized with estimates or probabilities detailed in Materials and Methods and in configuration files provided with the SIMCoV code in https://github.com/AdaptiveComputationLab/simcov.

Fig 2 shows the results of a small run of SIMCoV representing infection spreading over a 15 mm x 15 mm x 5 $\mu$m layer of lung epithelial tissue (9 million cells). The top row (Fig 2A) shows the state of epithelial cells (left), virion concentration (center), and inflammatory signal (right) at six days post infection (dpi), and the second row (Fig 2B) shows the same including extravasated T cells at eight dpi. The dynamics of the epithelial cells, the count of virions, and the number of T cells moving through the tissue are shown in the bottom row (Fig 2C). The

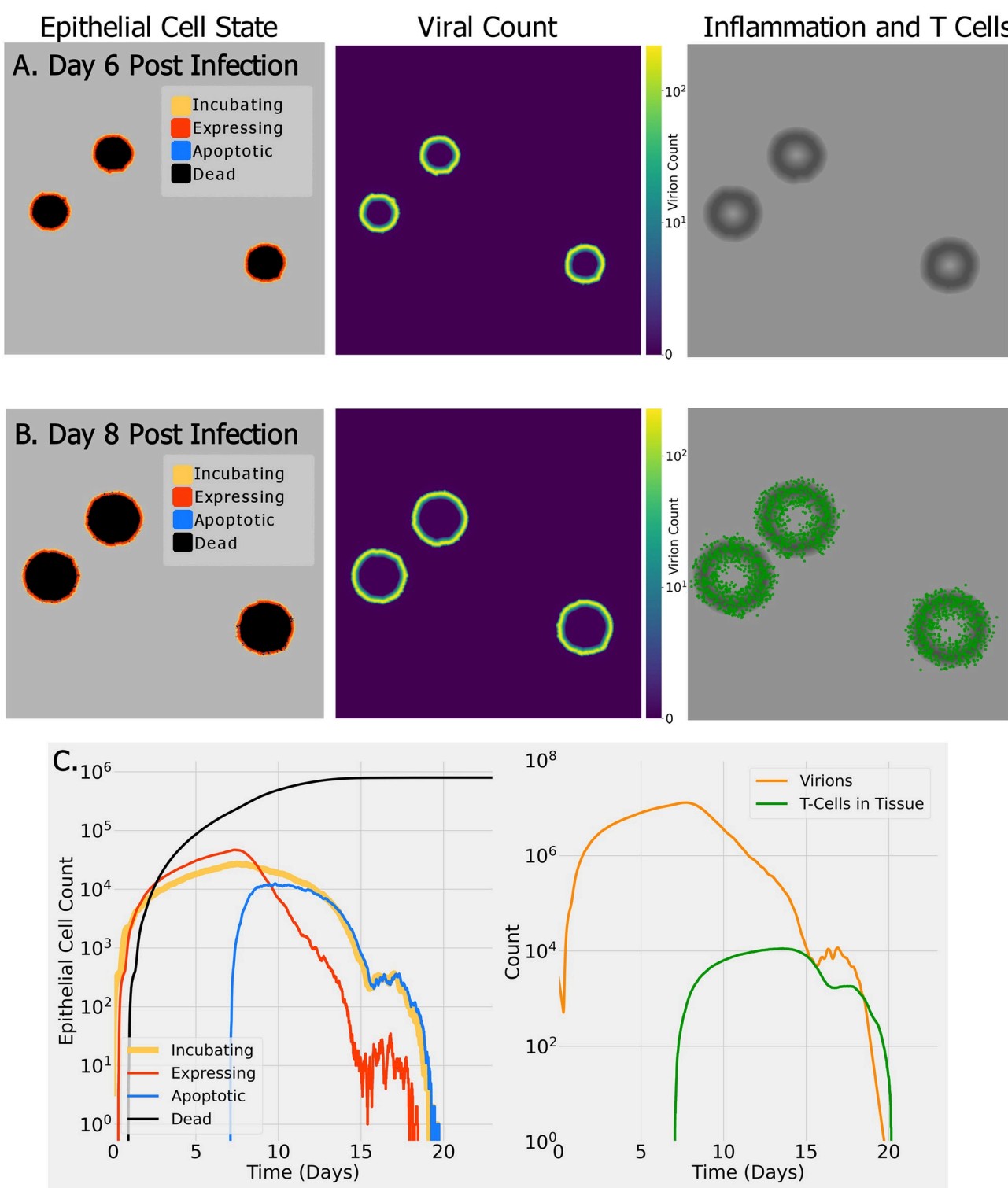

**Fig 2. Example SIMCoV simulation on a small 15 x 15 mm$^2$ layer of epithelial tissue.** (A) six days post infection (dpi); (B) eight dpi. Color indicates epithelial cell state (see legend, left column), virus concentration (see scale bar, middle column), and presence of inflammatory signal (in grayscale with darker regions having a stronger signal, right column) and T cells (green); (C) (left panel) shows the dynamics of epithelial cell state, and (right panel) shows virus and T cell dynamics. A video of this simulation can be viewed at https://youtu.be/e3fqe52--2k.

number of epithelial cells incubating and expressing virus (Fig 2C, left) and viral load (Fig 2C, right, orange line) all rise quickly after infection and then increase more slowly until seven dpi. The CD8[+] T cell response begins at seven dpi, and when T cells encounter infected cells, the infected cells become apoptotic. This reduces both the the number of incubating and expressing epithelial cells and the viral load, and the cumulative number of dead cells levels off. In this example, by approximately 15 dpi, the T cells have induced apoptosis in nearly all expressing epithelial cells, which leads to a reduction in produced inflammatory signals and consequently fewer T cells entering the lung. Then, the virus level begins to rebound. However, once incubating and newly infected cells express more virus and inflammatory signals, more T cells enter the tissue, encounter these expressing cells, and reinstate control over the infection.

## Peak viral load is proportional to the number of initial FOI

We used SIMCoV to test how spatially differentiated locations of initial infection affect viral dynamics. Images of the lungs of SARS-CoV-2 patients and experimental animal models suggest that there may be multiple sites of viral infection [28–30]. We tested the effect of seeding SIMCoV with multiple FOI on viral dynamics. We initialized simulations with one, four, and 16 spatially dispersed FOI (Fig 3A, square size not to scale). We found that each four-fold increase in the number of FOI increases both the peak viral load (Fig 3C) and number of extravasated T cells (Fig 3D) by a factor of approximately four (Fig 3B). This remains true when the absolute number of initially infected cells is held constant across all experiments, i.e., by creating a single FOI with a cluster of 16 adjacent infected cells and four FOI with four adjacent infected cells, so that all three experiments contain 16 initially infected cells but with varying number of FOI (Fig 3C). The initial number of infected cells only affects the peak viral load if those initial infections are dispersed across multiple FOI; peak viral load depends on the number of FOI, not the number of infected cells within each initial FOI. Virus diffuses from each spatially dispersed FOI independently, and with a larger number of FOI, there are more infectable cells available for virus to contact and infect. Greater spatial dispersion of infected cells also results in more inflammatory signal produced at more sites of infection, and T cells respond by entering the tissue in greater numbers, as seen in the correspondence between T cells and virions in (Fig 3C and 3D).

The replicated runs shown in Fig 3 are highly consistent for a fixed number of FOI for several reasons. Because the simulated tissue size is large relative to the FOI, even at their peak extent, it is statistically unlikely that randomly placed FOI will grow to overlap in the simulations. In Fig 3, each FOI, even at its peak extent, comprises less than 0.2% of the entire simulated tissue in our default configuration that represents 225 million cells. In addition, we located all FOI in the inner 80% of the simulated tissue to avoid boundary effects.

## Effect of CD8[+] T cell response on viral clearance

CD8[+] T cells that kill virally infected cells are another key factor that varies among patients [19, 22]. The CD8[+] T cell response depends on interactions in physical space: CD8[+] T cells are activated in lymph nodes, circulate in the blood, and migrate to the lung, where they extravasate into lung tissue at locations with sufficient inflammatory signal. After extravasation, they move through infected tissue until they directly interact with infected cells and cause cell death [31–34]. T cells likely play an important role in controlling SARS-CoV-2 infection; elderly individuals and patients suffering from severe COVID-19 disease have fewer circulating CD8[+] T cells [19, 35]. However, because it is difficult to study T cell responses in the lungs of infected patients, relatively little is known about how the presence of T cells in infected lung tissue might impact viral dynamics and control.

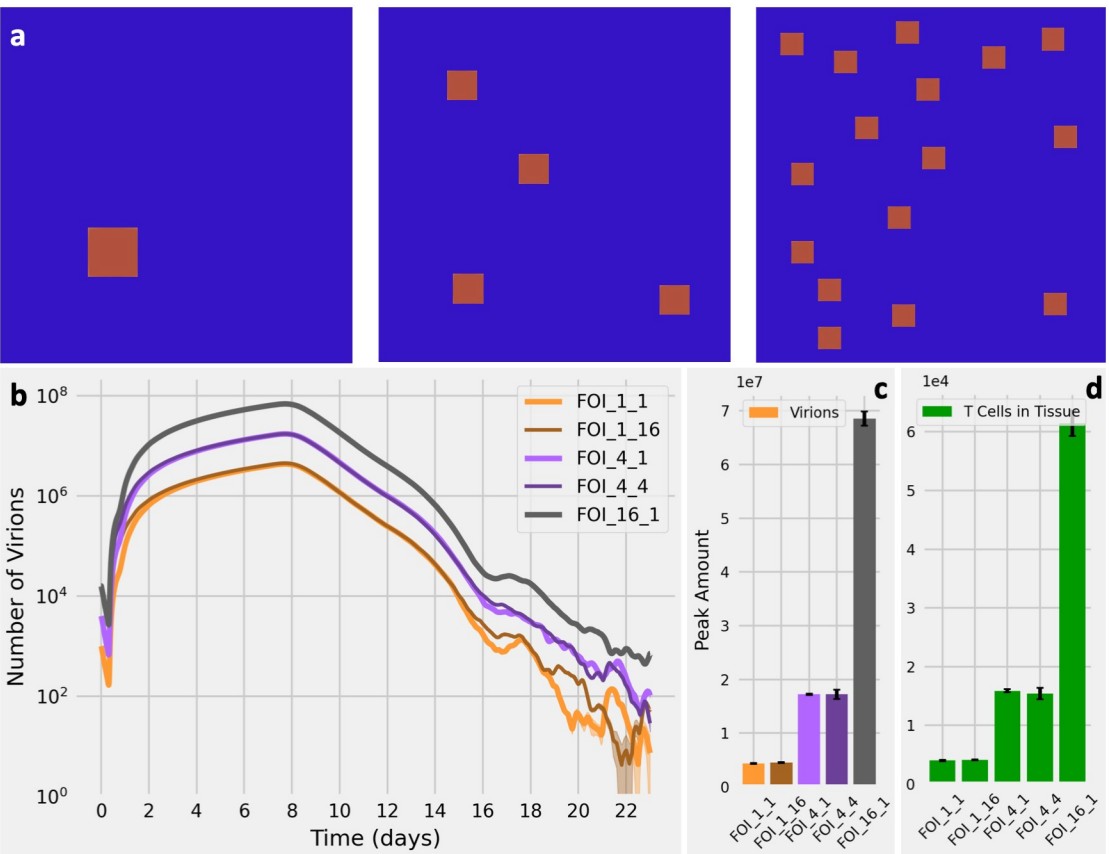

**Fig 3. Effect of the number of FOI on virus and T cell dynamics.** (A) Initial spatial distribution in SIMCoV for one FOI (left), four FOI (center), and 16 FOI (right). The area of each FOI is small in the simulation, but the squares shown are amplified relative to the simulation area in this image in order to make the relative sizes of the three cases visibly obvious. (B) Viral load over time given different FOI. The light orange line labelled FOI_1_1 represents one FOI initialized with a single infected cell, and the dark orange line labelled FOI_1_16 represents one FOI initialized with a cluster of 16 infected cells. Similarly, the purple lines represent four FOI, each initialized with one or four infected cells; the black line represents 16 FOI, each with one infected cell; 30 replicates with shading showing 95% CI (which is only visible at low numbers of virions). (C) Peak number of virions for different FOI with error bars showing the standard deviation for 30 replicates. There is a 3.90-fold increase from the one FOI configuration to the four FOI configuration and a further 3.99-fold increase in the 16 FOI configuration. These differences are statistically different ($p < 0.01$). However, the peak virions given FOI_1_1 and FOI_1_16 are indistinguishable ($p = 0.68$), as are the FOI_4_1 and FOI_4_4 cases ($p = 0.51$). (D) Peak number of T cells that extravasate into lung tissue for the different FOI. Similar to panel (C), the increase is 3.95-fold for each 4-fold increase in FOI.

We used SIMCoV to test how the magnitude of the immune response affects viral control. We found that varying the number of circulating T cells in the blood dramatically affects viral control (Fig 4A). Even a two-fold decrease in the production rate of circulating T cells can delay the timing of viral control by several days (compare 100,000 to 200,000 T cells/minute) or even change the outcome from viral control to viral persistence (compare 100,000 to 50,000 T cells/minute).

When the arrival of CD8+ T cells in infected tissue is delayed, the time it takes to reach peak viral load is also delayed, which allows a slightly higher peak load to be reached (Fig 4B). Delayed T cell arrival also increases the time until the infection is controlled, but it does not dramatically change the rate at which viral load is reduced. Even when T cell arrival is delayed from five to 15 days, viral control can eventually be achieved, albeit later in the infection. A

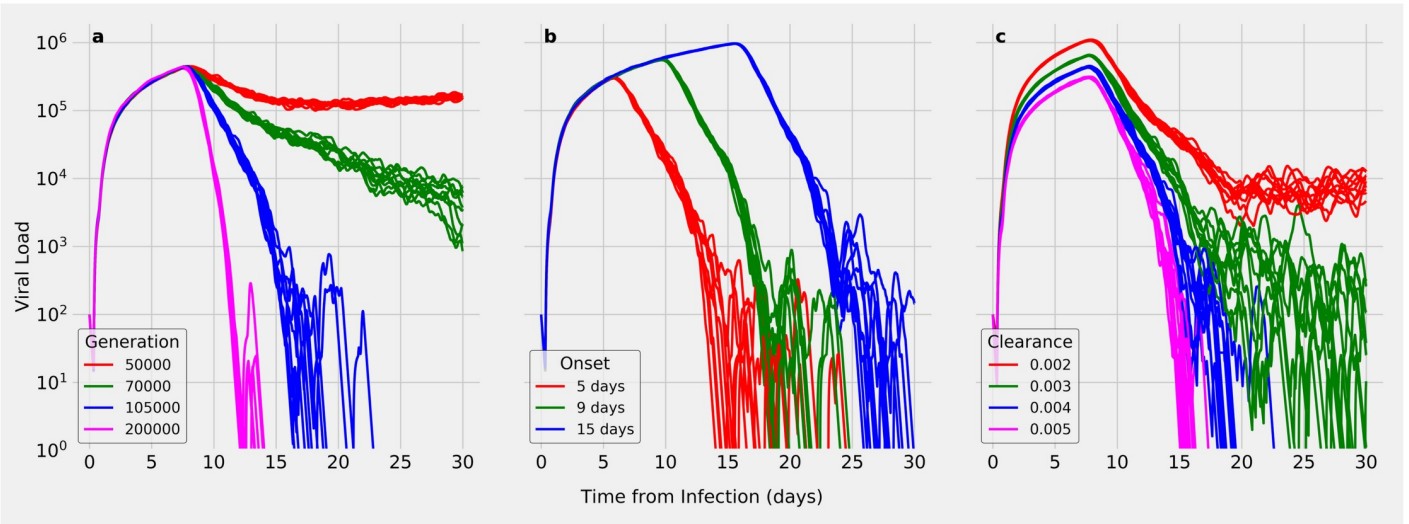

**Fig 4. Effect of varying key immune response simulation parameters.** Each subplot varies one parameter while holding others at the default values (Table 2) and shows the results of 10 random runs. (A) Varying T cell generation rate from 50,000 to 200,000 T cells produced per minute; (B) varying T cell delay; (C) varying viral clearance rate.

delay in viral control allows a longer period of elevated viral load, which may increase both the window of transmission and the extent of tissue damage, possibly leading to more severe disease.

The viral clearance rate is a parameter that captures the strength of the innate immune response, and, as can be seen in Fig 4C, changes in the viral clearance rate have a small effect on peak viral load and a much larger effect on viral dynamics in the later days of the infection. The magnitude of the innate immune response can make the difference between rapid viral control versus viral persistence leading to an infection that drags on indefinitely.

We performed a more complete analysis of how peak viral load and the percentage of infected cells vary given variations in individual parameters. We find that increasing viral production, viral diffusion, and the delay in T cell arrival into the lung each cause an approximately linear increase in viral load and percent of infected cells (Fig 5). The number of FOI has a particularly strong linear correlation with viral load and percent of infected cells. Our sensitivity analysis also confirms our findings in Fig 3 showing no effect of the number of virions within a single FOI. Similar to results in Fig 4, we find that varying parameters that affect the number of T cells in infected tissue, including T cell production, T cell lifespan in the tissue, and the inflammatory signal responsible for recruiting T cells into lung tissue show a threshold effect. That is, beyond a certain threshold, the viral load and percentage of infected cells are not sensitive to changes in these parameters. Our results suggest that below a certain number of CD8+ T cells, infection control is not achieved.

## SIMCoV viral loads compared to patient data

Having shown how the number of initial FOI and the timing and number of responding CD8+ T cells affect viral dynamics, we next consider how these factors affect SIMCoV fits to time courses of viral load in COVID-19 patients. Following previous ODE models, e.g. [7, 8, 12, 14], we use data from sputum samples reported in Wolfel et. al. [36]. This study collected sputum from the lower respiratory tract, in contrast to most viral swabs that are collected from the

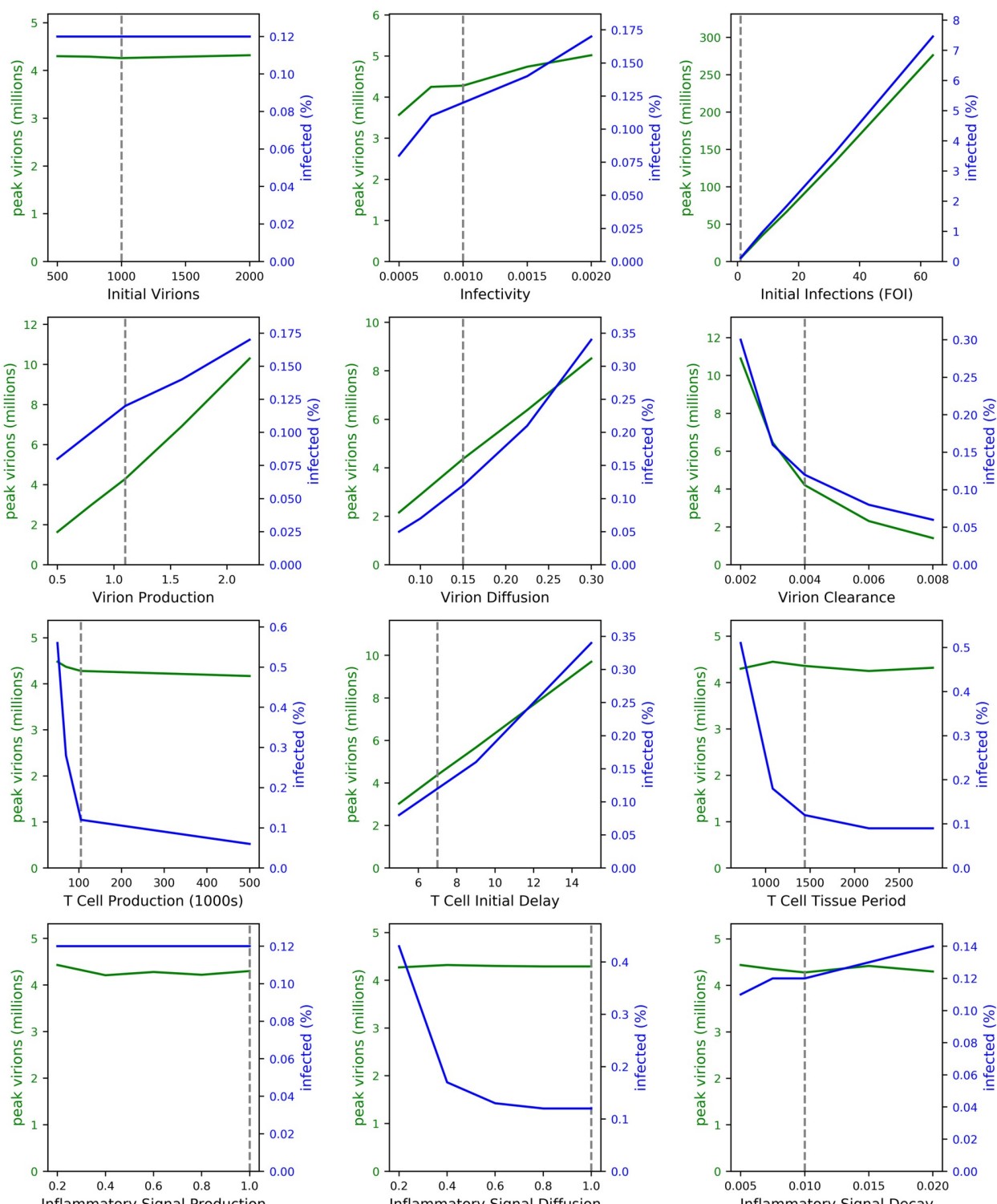

**Fig 5. The effect of varying model parameters, one at a time.** Each line is the average of five runs. Dashed lines indicate SIMCoV default values. The left axes (green lines) show how each parameter affects peak viral load and the right axes (blue lines) show the effect on the percent of cells in the simulation that become infected. The one-at-a-time sensitivity analysis is detailed in Materials and Methods.

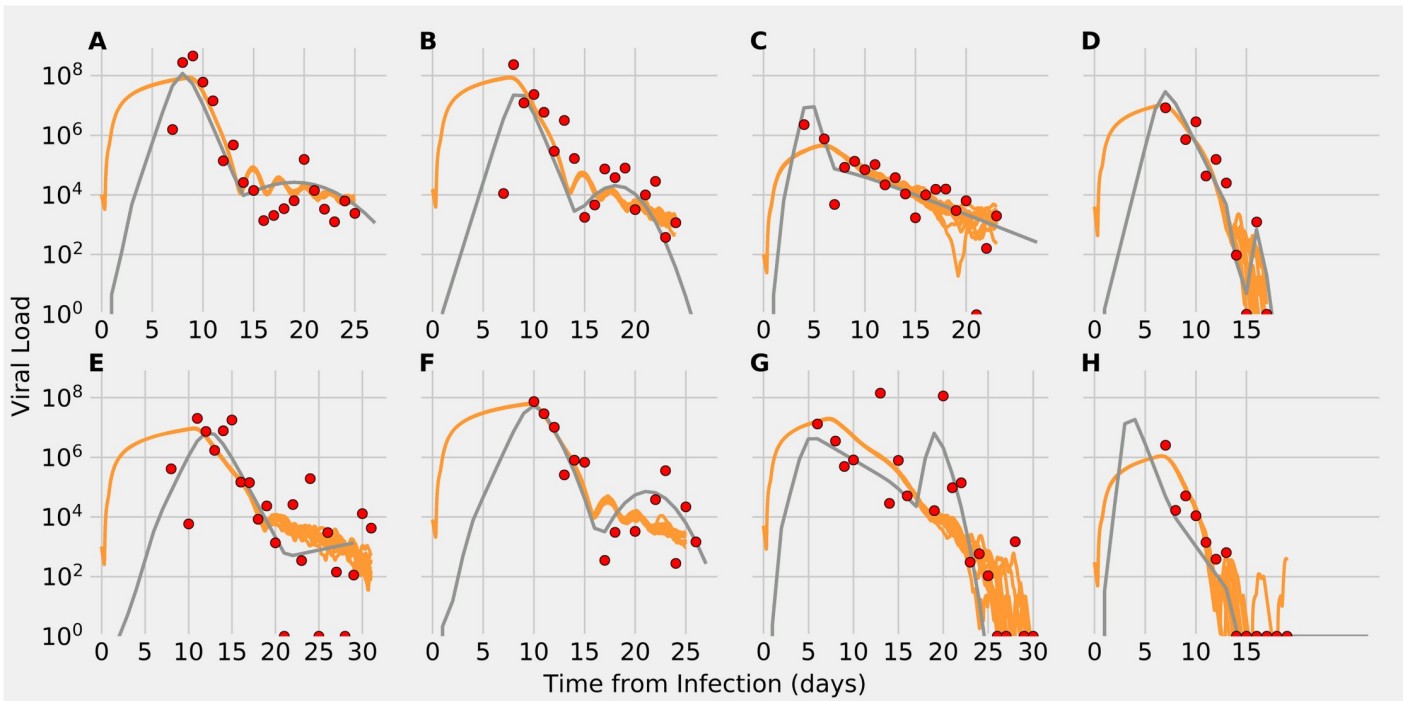

**Fig 6. Viral load predictions from SIMCoV simulations (orange) compared to the extended ODE in [12] (gray), and patient data from [36] (red points).** For each simulation, 10 random runs are shown.

nasal cavity. The lung contains vastly more tissue than the nasal cavity; the epithelial lining of the branching airways and alveoli forms a large and complex surface area of up to 80 m$^2$ [37]. Most earlier models of the Wolfel et al. patient data used a target cell limited (TCL) framework, in which viral load peaks when the virus infects all infectable "target" epithelial cells and runs out of targets. These models typically predict viral peaks within the first four or five dpi. However, SARS-CoV-2 peak viral load occurs later in the lung than in the upper respiratory tract (URT) [38], and the patient data have complex dynamics that last 15 to 30 dpi (Fig 6). Although the viral peak may be reached earlier in the URT, in the lung, it may not be achieved before the adaptive immune response begins [17]. Thus, TCL models may not be the most suitable approach for capturing infection dynamics in the lung, which is larger and has more complex organization.

We tested whether SIMCoV can explain the substantial variation in patient viral load data by varying only the initial FOI, timing and strength of T cell response, and viral clearance rates that include other immune activity. We began with default model parameters (Materials and Methods—Table 2 and Fig 2) and varied the number of FOI from one to 220 sites. We also varied the timing of first T cell arrival (from five to 10 days), T cell generation rate (from 90,000 to 200,000 cells per minute), and the viral clearance term (from 0.003 to 0.005) to account for variations in other components of the immune response.

We find that SIMCoV simulates the dynamics of patient viral load over the full 25-30 day time courses with fits that are quantitatively similar to those of the extended ODE described in Ke et al. [12] (Fig 6). Table 1 shows that the goodness-of-fit is similar between SIMCoV and the ODE as measured by the Root Mean Squared Log Error (RMSLE). SIMCoV was parameterized manually, as described in the subsection "Manually Parameterizing SIMCoV to Fit

**Table 1. SIMCoV parameter settings used for each patient; RMSLE for the simulation averaged over 10 runs and RMSLE for the ODE.**

| Patient | T Cell Generation Rate (1000 T cells/min) | T Cell Onset (days) | Initial Sites of Infection | Viral Clearance | SIMCoV RMSLE | ODE RMSLE |
|---|---|---|---|---|---|---|
| A | 200 | 8 | 220 | 0.003 | 1.98 | 1.79 |
| B | 160 | 7 | 150 | 0.003 | 2.58 | 2.55 |
| C | 90 | 5 | 1 | 0.003 | 2.31 | 2.05 |
| D | 120 | 6 | 40 | 0.005 | 2.65 | 1.62 |
| E | 120 | 10 | 10 | 0.003 | 3.75 | 3.40 |
| F | 150 | 9 | 80 | 0.003 | 2.30 | 1.92 |
| G | 90 | 6.5 | 50 | 0.004 | 3.54 | 2.88 |
| H | 160 | 6 | 3 | 0.004 | 2.30 | 1.82 |

Patient Data" of the Methods section. When we varied the initial FOI, SIMCoV replicated the several order-of-magnitude differences in viral load observed across the eight patients. Compared to the ODE model, SIMCoV produced a less pronounced peak with higher viral loads early in infection that persisted for several days. The SIMCoV curves are consistent with the observation that individuals infected with SARS-CoV-2 are most infectious early in infection before symptom onset [39, 40]. The SIMCoV runs also predict that elevated viral titers persist for more days, which agrees with patient viral load data.

Varying the T cell generation rate and first day of arrival captures the unique shapes of viral load trajectories after seven dpi. SIMCoV exhibits both oscillations in viral load and second peaks that are seen in some patients [36]. These results support the hypothesis that decreased T cell numbers can account for viral persistence and inability to control viral load in some individuals [19, 20]. The fit to Patient A also shows that a high initial and peak viral load, which we fit by setting the number of FOI to 220, is not quickly controlled by even a high T cell response (in this case, a T cell production rate of 200,000 T cells/min beginning on 8 dpi). This suggests that initial infection in a large number of FOI could overwhelm even a robust T cell response.

Previous TCL models have replicated the data in Wolfel et al. [36] by expanding the availability of target cells later in infection and varying many parameters in each patient [7, 12]. For example, the extended ODE model in [12] achieves good fits by varying infectivity, infected cell death rate, virion production rate, number and timing of new target cells introduced to represent newly seeded infections, and the exponential increase of the infected cell death rate to account for the adaptive immune response. By incorporating spatial interactions, SIMCoV provides a more parsimonious explanation for the dramatically different viral load trajectories among SARS-CoV-2 positive individuals by varying only the number of FOI in the initial infection and the magnitude and timing of the immune response, particularly the CD8$^+$ T cell response. These factors are consistent with hypothesized explanations for highly variable patient outcomes. Elderly individuals have lower T cell counts, and our model suggests that fewer T cells along with slower T cell response could help explain why elderly individuals have increased disease severity [19, 22]. More speculatively, the number of FOI might be larger if a larger amount of virus is inhaled into the lungs; a larger initial viral dose has been shown to increase disease severity in macaques [41] and has been suggested in humans [42, 43]. Variation in FOI and T cell response may also account for the long infectious period in some patients [4–6].

## Modeling viral spread over branching airways

The SIMCoV experiments described above simulate a single layer of epithelial cells (Figs 2–6). However, the lung is a highly complex structure of branching airways descending from large

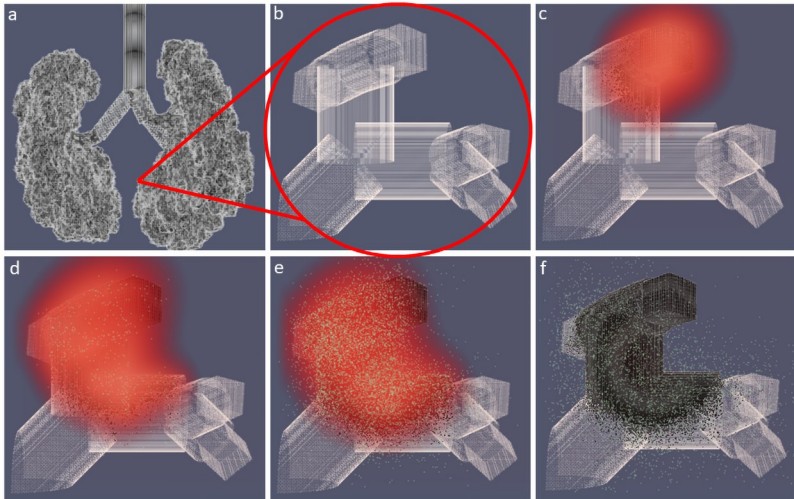

**Fig 7. Simulated 3D branching airway with a single FOI. (A)** shows the full lung model; **(B)–(F)** show T cells (green), virions (red), and a small region of epithelial cells (gray) at **(B)** 0 dpi, **(C)** 4.2 dpi, **(D)** 6.3 dpi, **(E)** 6.9 dpi, and **(F)** 8.3 dpi. Dead epithelial cells are indicated in black. A video corresponding to these images is available at https://youtu.be/ychAmDU4qFM.

bronchioles down to alveoli lined by epithelial cells, with much of the volume consumed by alveolar space for gas exchange [37]. To investigate how the lung airway structure impacts viral spread and T cell control, we developed a bifurcating fractal branching algorithm and integrated it into SIMCoV simulations to form the surface of the epithelial layer. Following Yeh et al. [44], we model 26 branching generations (from the trachea to alveolar sacs) in all five lobes of the lung. The 3D branching model incorporates trachea, bronchi, bronchioles, and ducts, represented as cylinders of epithelial cells with length, diameter, branching angle, and gravity angle calculated at each level of the branching hierarchy, and each terminal alveolar sac is represented as a hollow 200 x 200 x 200 $\mu$m$^3$ cube of epithelial cells. Fig 7A shows the full model of the branching structure, and Fig 7B–7F show how virus spreads over a small region of bronchioles and alveoli in a 1500 x 1500 x 1500 $\mu$m$^3$ volume.

Fig 8 shows how different topologies of epithelial cells modeled in SIMCoV affect the predicted rate of viral spread in the first seven dpi and before the T cell response. The default model parameters are used for all topologies, and each simulation was run with sufficient numbers of epithelial cells to avoid reaching target cell limitation. The branching network model (Fig 7) best captures the spatial arrangement of epithelial cells. However, viral spread over the branching network (blue line in Fig 8) shows only four times greater growth than spread over the simple 2D layer (yellow line) used in our earlier simulations. In contrast, an undifferentiated 3D grid of cells (red line) generates 66 times more infected cells than the branching model at the end of the first week of infection, and the TCL ODE from [12] (green line) generates thousands of times more infected cells before reaching target cell limitation within the first week.

We hypothesize that the 2D model slightly underestimates viral growth because it is constrained by the circumference of the circle of infection, which grows linearly with the total number of cells in the FOI (Fig 2). In (non-branching) 3D grid simulations, infection spreads from the faster growing sphere of infected cells. ODE models assume all infected cells can infect all other infectable cells, and without spatial constraints, the virus spreads even faster. At least at the scale of the hundreds of millions of cells modeled here, the simple 2D model is

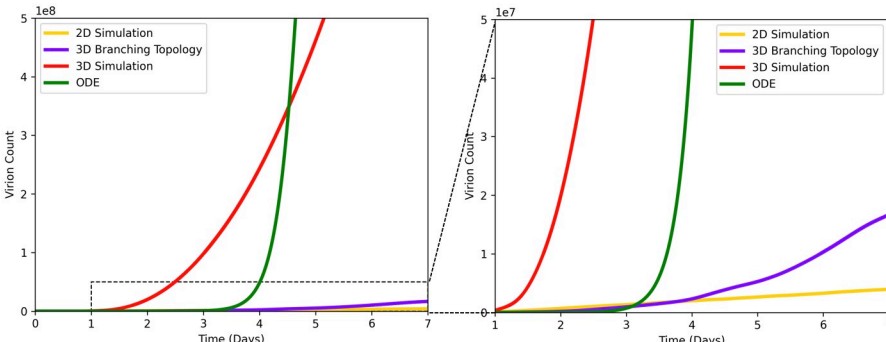

**Fig 8. Viral spread from a single FOI given different model topologies.** Virion count is plotted vs. time for four different SIMCoV configurations: 2D grid (yellow); 3D grid (red); 3D branching model (blue); ODE model (green). The right panel is zoomed in on the y-axis (showing up to $10^7$ virions) to show the dotted region of the left panel (which shows up to $10^8$ virions).

computationally efficient and captures the dynamics of the branching model well (although small differences are visible in the right panel of Fig 8). Future work can incorporate more detailed movement of virus and immune cells through the epithelium and branching airways as well as the percent and locations of infectable cells in simulations at the full scale of the human lung.

# Discussion

## Summary

SIMCoV models the spread of SARS-CoV-2 through the lung and subsequent immune control by CD8[+] T cells. There are three main findings from our simulations. First, variation in the initial number of sites of infection (or FOI) can dramatically change the peak viral load. Second, variation in the magnitude and timing of the CD8[+] T cell response can explain why some patients clear infections quickly while viral load oscillates and lingers in others. Third, modeling the spatial spread in a 2D layer of tissue, or more realistically, in a fractal branching model of airway epithelial tissue, results in much slower viral spread than in mathematical models that do not consider spatial constraints.

Fig 3 shows that the number of initial FOI determines peak viral load, but the actual number of virions causing initial infection does not independently affect the viral peak. However, in real infections, the likelihood of virus taking hold in more FOI might increase if more virus is inhaled. The number of inhaled virions and their spatial dispersion may be a key explanation for why viral load and disease outcome can be so different in demographically similar patients. An important insight from SIMCoV is that spatial dispersion of virus may be particularly important for SARS-CoV-2 and other lung infections because the epithelial surface area of the lung is so large; this contrasts with the smaller surface areas of the nasopharynx, where TCL can be reached quickly even with a small initial infection [17].

Fig 6 shows that varying T cell arrival time by only a few days (from five to 10 dpi, consistent with [45]) and the magnitude of the T cell and other immune responses by two-fold accounts for the variability and oscillations in viral clearance of the patients from [36]. This is consistent with studies that show CD8[+] T cells in particular are critical for clearance and control of SARS-CoV-2 infection [18, 19, 22, 46, 47].

Fig 8 shows how the branching topology of the lung (shown in Fig 7) impacts viral spread compared to a simple 2D or 3D grid and a non-spatial ODE model. The 2D model (which we used to simulate viral dynamics in Figs 2–6) produces fits most similar to the more realistic branching airway model. In contrast, the 3D and ODE models both predict much faster viral growth than the branching model. ODE models assume well-mixed (also called "mass action") interactions in which any infected cell can infect any other infectable cell regardless of the physical distance between them. ODE models with TCL require extremely fast viral replication (e.g., $R_0 \approx 27$ in [12]) to reach TCL and fit the timing of the viral peak.

The relatively slow viral spread over surfaces (either the branching airway surface or the simpler 2D layer of cells) means that it is unlikely that TCL can explain the peak viral load. Our hypothesis that not all lung cells are infected is consistent with observed mild illness in many patients, including those in [36]. It is also consistent with CT scans showing damage in only a small percentage of tissue in most patients [28, 48]. Because SIMCoV includes spatial constraints on viral spread, it suggests that immune response, particularly CD8$^+$ T cell response, determines the timing of the peak viral load, rather than the time at which virus infects all infectable cells.

## Relationship to other models

Several earlier SARS-CoV-2 ODE models were tuned to fit the same lung viral load data from [36] that we fit here. Some included an implicit immune response incorporated in the cell clearance term and an explicit adaptive immune response, e.g. [8, 14–16, 49], similar to the approach we took with SIMCoV. SIMCoV complements these earlier ODE models but makes different predictions. For example, earlier models predict a sharp early peak in viral load in the first few days of infection, while SIMCoV predicts a broader peak which rises rapidly for the first few days post infection and then continues to increase slowly until the T cell response begins between day seven and 10 [45]. Viral levels then decline gradually but continue to oscillate as infected cells continue to produce virus and as T cells discover pockets of hidden infection. The viral dynamics predicted by SIMCoV are consistent with observations that the viral load in the lung peaks later than in the upper respiratory tract and can persist for much longer [16, 17, 50]. In contrast to ODE models, SIMCoV suggests a parsimonious, mechanistic explanation that peak viral load in the lung is determined by the number of FOI and the magnitude and timing of the CD8$^+$ T cell response.

Other spatially explicitly models of viral spread through the lung have been developed. SIMCoV replicates many of the features of the CyCells ABM that modeled the spatial spread of tuberculosis [26] and influenza [25] over a 2D grid. Sego et al. [51] built a multi-scale model of SARS-CoV-2 that connects a 2D spatial grid that, like SIMCoV and CyCells, represents epithelial cells as a static 2D layer with virus and signals diffusing as fields over that grid. Quirouette et al. [52] used partial differential equations to model diffusion and advection of influenza along the single dimension from the lower to upper regions of the lung. Each of these attempts to model spatial dynamics has strengths and limitations.

SIMCoV is unique in its ability to explicitly model the spread of virus from multiple FOI as well as the recruitment of T cells to those locations. To do this, SIMCoV uses a highly scalable architecture to represent hundreds of millions of cells that can contain multiple growing FOI. T cell killing of infected cells requires direct cell to cell contact [31, 33, 53]. Because SIMCoV represents spatial positioning and T cell movement explicitly, their physical co-location with infected cells arises naturally in the model. This modeling strategy provides a mechanistic explanation that shows how even a small decrease in the number of responding T cells changes viral dynamics from a smooth decline to an oscillating decline to uncontrolled viral

persistence. Interestingly, later arrival of T cells into the infected lung delays viral control but otherwise has little effect on viral dynamics or clearance.

## Caveats, limitations, and future work

SIMCoV, like any model, focuses on only a subset of possible factors. In this paper, we focused on the CD8+ T cell response and not innate, antibody, B cell, or CD4+ T cell immunity, and we focused on spread through the lung and not the nasal cavity. Some parameters in the model are not well characterized in the biological literature, i.e. rates of production, diffusion and decay of inflammatory signals. Fig 5 shows that peak viral load and the extent of infection are largely insensitive to these parameters.

When fitting the model to patient viral loads, we kept parameters related to virus characteristics (i.e. viral infectivity and production rates) constant because we do not expect viral characteristics to vary across patients. However, these parameters might vary significantly between different SARS-CoV-2 variants. A logical avenue for future work is to better understand the dynamics of different SARS-CoV-2 variants, particularly Delta and Omicron, by modeling its properties, such as increased viral production and increased infectivity. Future studies could also explicitly represent antibody responses and CD4+ T cell responses, either as time varying components of the clearance term or by modeling their ability to neutralize virus or reduce viral replication or entry into cells. This would enable comparisons of naive patients with vaccinated and previously infected patients. Lipsitch et al. speculate that a higher viral inoculum might reduce vaccine efficacy [54]; future SIMCoV experiments can assess how the number of FOI impact T cell control of infection in vaccinated people. SIMCoV could also be used to study other respiratory diseases, such as influenza, or even be extended to model infection in other organs. The SIMCoV platform could be used to simulate other spatial interactions, like predator prey dynamics between immune and infected cells or collective action dynamics [11], such as the collective search strategies of T cells [55] or the movement patterns of T cells within the lung [32, 34].

Previous spatial models were limited both by data on infection dynamics and the computational power required to explicitly represent spatial interactions among large numbers of cells [23]. SIMCoV overcomes these limitations by leveraging data shared in response to the COVID-19 pandemic and a model design and implementation that takes advantage of HPC capabilities. To enable open and reproducible science, SIMCoV is freely available under an open-source license, and it was designed to be easily extensible. Future extensions of the model could scale up to a full lung, investigate the effects of mucus and the complex dynamics of airflow in the respiratory tract [56], and incorporate more detailed topological models of the airways and alveoli, including the fraction and distribution of epithelial cells with receptors susceptible to infection. By predicting the time course of viral loads within individuals, SIMCoV can help to identify factors that determine windows of transmission between individuals and thereby improve understanding of epidemic spread.

## Materials and methods

### Model description

SIMCoV is an Agent Based Model (ABM). The SIMCoV source code is available at https://github.com/AdaptiveComputationLab/simcov. Space is represented by a discrete Cartesian grid that can be either two or three-dimensional. For clarity, we describe the model with reference to a 2D layer of epithelial cells, although the mechanisms apply to other topologies, in particular, a 3D grid and a 3D branching structure of lung airways.

Grid points are spaced five microns apart (roughly the diameter of a T cell), and components of the model occur only at these discrete locations. The model is run as a discrete-time simulation, where each time step represents one minute, approximately the time it takes for a T cell to move five microns (one grid point) [34, 57]. At each time step, the model components are updated according to their states and interactions. Grid boundaries are fixed, not periodic.

There are four main components of SIMCoV: epithelial cells, CD8$^+$ T cells, virions, and inflammatory signals, representing the subset of cytokines that cause T cell extravasation into the lung tissue. The interactions between these components are shown in Fig 1. Each epithelial cell has a fixed grid location with no more than one cell per location. However, some locations can be empty, representing airways (as can be seen in Fig 7). In addition, a single T cell may also occupy a grid point, and every grid point has a concentration of virions and inflammatory signals. The number of virions per grid point is represented explicitly but capped at 125,000 virions, which is the maximum number estimated to fit within 125 $\mu$m$^3$. The concentration of inflammatory signal is represented as a floating point number between $1 \times 10^{-6}$ and one.

Each simulation run begins with a number of sites of infection (parameter **b** in Table 2), which are grid points with an initial number of virions (parameter **c**). At each time step, the virions diffuse uniformly at a given rate (parameter **j**) to neighboring grid points, both those with and without epithelial cells, and some fraction (parameter **i**) of the virions are cleared, reducing the concentration.

Initially, all epithelial cells are *healthy* (uninfected) and susceptible to infection. At the end of each time step, if there are virions in the same location as a healthy epithelial cell, that cell becomes infected (transition **1** in Fig 1) with probability proportional to the virion concentration multiplied by *infectivity* (parameter **g**). An infected epithelial cell is initially *incubating* and remains in this state for a number of time steps, which is sampled from a Poisson distribution (parameter **d**). An *expressing* epithelial cell produces virions at a constant rate per time step (parameter **h**, transition **3a**), which are added to the local concentration and diffuse

**Table 2. Table of SIMCoV parameters.** Default values are COVID-19 parameters assembled from a variety of sources. Time periods are Poisson distributions, defined by the mean, λ. Each time step is one minute, and grid points are five microns apart. Unless stated otherwise, the default parameters were used for all experiments. Parameters marked with (*) were not taken from the literature—see Parameter Derivation.

| Parameter | Description | Default | Reference |
|---|---|---|---|
| (a) Dimensions | Simulation size in x, y, and z dimensions | 15000x15000x1 | * |
| (b) Initial Infections | Number of grid points with initial infections | 1 | * |
| (c) Initial Virions | Number of virions at initial infection locations | 1000 | * |
| (d) Incubation Period | Average minutes until an infected cell starts expressing virions | 480 | [58] |
| (e) Expressing Period | Average minutes after expressing starts until cell death | 900 | [10] |
| (f) Apoptosis Period | Average minutes after apoptosis is induced until cell death | 180 | [59] |
| (g) Infectivity | Probability of one virion infecting one cell per minute | 0.001 | [25] |
| (h) Virion Production | Number of virions produced by expressing cell per minute | 1.1 | [60] |
| (i) Virion Clearance | Fraction by which virion count drops per minute | 0.004 | [25] |
| (j) Virion Diffusion | Fraction of virions that diffuse into all neighboring grid points per minute | 0.15 | [25] |
| (k) Inflammatory Signal Production | Concentration of inflammatory signal produced by expressing cells per minute | 1 | * |
| (l) Inflammatory Signal Decay | Fraction by which inflammatory signal concentration drops per minute | 0.01 | * |
| (m) Inflammatory Signal Diffusion | Fraction of inflammatory signal that diffuses into all neighboring grid points per minute | 1 | [25] |
| (n) T Cell Production | Number of T cells generated per minute into circulation | 105000 | [25, 61] |
| (o) T Cell Initial Delay | Average minutes before T cells start to be produced | 10080 | [45, 62, 63] |
| (p) T Cell Vascular Period | Average minutes before death for a T cell in the vasculature | 5760 | [25] |
| (q) T Cell Tissue Period | Average minutes before death for a T cell after it extravasates | 1440 | [64] |
| (r) T Cell Binding Period | Average minutes T cell is bound to an epithelial cell | 10 | [33] |

(transition **4a**) until they are cleared (transition **4b**). Expressing cells that avoid contact with a T cell eventually die (parameter **e**, transition **3c**).

When epithelial cells are *expressing* (and *apoptotic*, described below), they produce inflammatory signal at a constant rate (parameter **k**) per time step (transition **3b**). Those signals diffuse to neighboring grid points and decay at a constant rate (parameters **m** and **l**, respectively). Inflammatory signal is represented as a floating point number (concentration) with a minimum value ($1 \times 10^{-6}$), below which it is considered eliminated. In the current implementation, inflammatory signals only affect T cell extravasation, although future extensions could incorporate other chemokines and cytokines.

After some delay (parameter **o**), activated antigen-specific CD8$^+$ T cells enter the blood from (unmodeled) lymph nodes (transition **7**) at a constant rate (parameter **n**). T cells in the vasculature extravasate when they detect inflammatory signal (transition **9b**). T cell movement through the vasculature is not explicitly modeled; instead, a random location is chosen for each T cell at every time step, and if that location contains inflammatory signal, then the T cell extravasates. The random location is chosen from among all of the possible locations for a complete lung, but when only a fraction of the lung is being modeled, the probability of extravasation is reduced accordingly. Since only one T cell can occupy any single location in the tissue at a time, T cells are blocked from extravasating if the location already contains a T cell. T cells that do not extravasate continue to circulate (transition **9a**); those that fail to extravasate before the vascular T cell lifespan (parameter **p**) die and are removed from the simulation (transition **9c**).

T cells residing in lung tissue can bind to epithelial cells at the same location or at an immediate neighboring location (transition **10a**). Binding probability is proportional to how long the cell has been incubating, with a maximum of one if the cell is expressing. Multiple T cells can bind to a single infected cell (respecting the occupancy rules outlined earlier). When binding occurs, the T cell transitions to *apoptotic* (transition **3d**) and after some time (parameter **f**), it dies (transition **6b**). Apoptotic epithelial cells produce virions (transition **6a**) and inflammatory signals at the same rates as expressing cells. T cells remain bound for a fixed time (parameter **r**), after which they continue moving in the tissue. During each time step, the T cell moves one step to a randomly selected adjacent location that is not occupied by another T cell. Hence, T cells move at a maximum rate of five microns per minute [34]. After moving, if a T cell is adjacent to or colocated with an infected cell, it can bind again. This process of random movement and possible binding continues until the T cell dies (transition **10b**, parameter **q**). T cells residing in lung tissue never return to the vasculature.

## Parameter derivation

Where possible, parameter settings were derived from published data, as described below:

a. *Dimensions*: In general, each dimension is set large enough to avoid target cell limitation during the first week post-infection; for the default 2D simulation, the dimensions are 15,000 x 15,000 x 1, representing a 75mm x 75mm layer of 225 million cells.

b. *Initial Infections*: Set to one to represent the minimum number of infections to simulate viral spread.

c. *Initial Virions*: Set according to *infectivity* (**g**), so as to ensure a successful initial infection of at least one cell.

d. *Incubation Period*: Roughly, the eclipse period; estimated to be 7-8 hours or 480 time steps in [58].

e.  *Expressing Period*: Estimated to be about 15 hours or 900 minutes in [10], which is similar to the expressing period of the influenza virus (about 1000 minutes [25]).

f.  *Apoptosis Period*: Estimated to range between 48 minutes and 168 minutes in mice [59]; we estimate a slightly longer period, 180 minutes, for humans.

g.  *Infectivity*: Set to 0.001, which is the probability of infection per cell per minute, i.e. it will take on average 1,000 minutes for a virion to infect a single cell. This is within the range of 12 to 1,200 minutes estimated for influenza [25], although this value is largely unknown and almost certainly varies between diseases and viral variants.

h.  *Virion Production*: Calculated by dividing the burst size by the expressing period (**e**); the burst size is the total number of virions produced by an infected cell, which was found to be 1,000 virions [60].

i.  *Virion Clearance*: Set to 0.004 virions per minute, which is within prior calculations of the virion decay rate of the influenza virus, ranging from 7e-7 to 0.07 virions per minute [25]. We vary this parameter from 0.003 to 0.005 to fit patient data.

j.  *Virion Diffusion*: Calculated from the Einstein-Smoluchowski equation for particle movement, $D \approx \alpha x^2/(2t)$, where $x$ is the mean distance diffused in one direction along one axis in time $t$, and $\alpha$ is the SIMCoV parameter. Hence, for a mean distance of five microns (one grid point), a time of one minute (one simulation time step), and a virion diffusion parameter of 0.15, the diffusion coefficient is $0.0312\mu m^2/s$. This diffusion coefficient is similar to the diffusion coefficient calculated for influenza [25], which is $0.0318\mu m^2/s$.

k.  *Inflammatory Signal Production*: The concentration of inflammatory signal produced per minute is not a value explicitly available from the literature because it is an abstraction of the production rates of multiple signaling molecules, each of which varies for different pathogens. For example, in [25, 27] production of CXCL10 (IP-10) and CCL5 (RANTES) vary by several orders of magnitude for different strains of influenza. Therefore, we model inflammatory signals as an abstract quantity with a production rate of 1 unit per time step. The effect of this parameter on T cell extravasation then depends on inflammatory signal decay and diffusion parameters. The inflammatory production, decay, and diffusion parameters result in an area or volume containing inflammatory signals that is two to three times that of infected cells in simulations with default parameters.

l.  *Inflammatory Signal Decay*: We estimate a half life of 70 minutes, which is similar to the half life of 30 minutes estimated in earlier work for influenza [25].

m.  *Inflammatory Signal Diffusion*: Calculated from the Einstein-Smoluchowski equation for particle movement, $D \approx \alpha.x^2/(2t)$, where $x$ is the mean distance diffused in one direction along one axis in time $t$, and $\alpha$ is the SIMCoV parameter. Hence, for a mean distance of five microns (one grid point), a time of one minute, and an inflammatory signal diffusion parameter of 1, the diffusion coefficient is $0.210\mu m^2/s$, which is within the range of the chemokine diffusion coefficient calculated in [25] (from $0.00318\mu m^2/s$ to $318\mu m^2/s$).

n.  *T Cell Production*: The number of T cells produced by replication in the mediastinal lymph nodes of the lung is calculated by scaling up the T cell production rate in Levin et al. [25], which was estimated from mouse models [61]. That previous calculation assumed that a single virus-specific T cell encounters an antigen-presenting cell and begins to replicate on day 0 post infection, doubling every eight hours. After five days,

those activated CD8$^+$ T cells could generate approximately 30,000 new T cells per day (or 21 per minute) into circulation. For SIMCoV, we assume a linear 5,000-fold increase from mouse to human, which reflects the approximate increase in lung and blood volume and gives a rate of 105,000 T cells per minute entering circulation.

o.   *T Cell Initial Delay*: Estimated from [45] for SARS-CoV-2. The SIMCoV default is seven dpi, which we vary from five to ten dpi to fit patient data. We note that T cell delay is related to the T cell production rate. If T cells double every eight hours and replication in humans lasts 2.3 days longer than in mice (seven versus five dpi), this would cause a 128-fold increase in T cell production. With 40 times greater SARS-CoV-2 specific T cell precursors, this would generate 5,000 times greater T cell production in humans. Alternatively, exponential replication of a single T cell precursor for nine days would generate 5,000 times greater production in humans compared to mice.

p.   *T Cell Vascular Period*: Equivalent to the T cell age in the blood used in a model of T cell responses to influenza [25].

q.   *T Cell Tissue Period*: Largely unknown, but estimates were provided in Keating et al. [64].

r.   *T Cell Binding*: Taken from Halle et al. [33].

Fig 5 explores the impact of varying model parameters. We used a one-at-a-time (OAT) approach for sensitivity analysis, similar to the method reported in Hoertel et al. [65]. OAT is useful when it is computationally expensive to run many simulations and when some parameters are predetermined or have a small acceptable range [66]. Thus, our analysis varied one parameter at a time while holding the others at their default values. We report the size of the peak viral load during the run and the percentage of infected cells at the end of the run (including dead cells). Several of the parameters have only a small impact on disease progression: *initial virions* (**b**), *T cell tissue period* (**q**), and *inflammatory signal production and decay* (**k, l**). Other parameters have clear and expected effects; for example, there is a linear relation between peak viral load and *virion production* (**h**).

## Manually parameterizing SIMCoV to fit patient data

We developed the fits to patient data manually, using an approach based on plausible biological mechanisms related to the immune response. We hypothesized that the differences among individual patients arose as a consequence of different immune responses and initial viral exposure, rather than from changes in the nature of the virus from one patient to the next. We also focused on adjusting the subset of model parameters that are expected to vary from patient to patient, depending on age, health, etc., and initial viral exposure. First, there are the parameters that control the extent of the immune response, including T cell production and onset delay and viral clearance rate. Second are the parameters that capture the size of the initial exposure: the number of FOI and initial virions parameters (the latter has no impact, as seen in Fig 5, so we ignored it in the fitting process). All other parameters were kept constant at the default values because we do not expect them to vary between patients. For example, several parameters define disease characteristics, such as virion production and diffusion and infectivity. In the future, these could be varied to explore the impact of different variants or different pathogens; we hypothesized that those factors are unlikely to vary substantially across patients in the Wolfel dataset [36].

In all cases, the four selected parameters were adjusted to obtain the fit as follows. We set the *T cell initial delay* according to the timing of peak viral load, using a value one day (1440 minutes) before the empirical peak. Second, we set the *initial infections* (the number of FOI) to obtain a matching peak viral load. Because the model produces a count of virions deep within

the lungs, and for the patients we only have sputum samples with counts of RNA copies, we do not expect a one-to-one mapping between empirical viral load and the model virion counts. To address this, we used a scaling factor of 0.1, chosen to minimize the number of FOI required for matching (and hence the computational cost). This scaling factor is the same for all patients; we match the peak scaled values by adjusting the number of FOI. As can be seen in Fig 5, there is a linear relationship between the number of FOI and the size of the peak viral load, so it is a simple matter to adjust the FOI to match the observed peak. As can be seen in Fig 4A, the T cell generation rate affects the shape of the curve after the T cell response begins. Hence, we adjusted this value to match the post-peak curve observed in the patient data. If the viral load continues to oscillate for some period after the initial drop (patients A, B, C, E, F), then we reduced the clearance rate from the default of 0.004 down to 0.003. This results in more "hidden" pockets of incubating virus that evade detection and cause repeated surges in the later stages of disease progression. Finally, there is one case (patient D) where the infection is cleared very rapidly, and in this case, we increased the clearance rate to 0.005.

## Model implementation

The SIMCoV model is implemented in UPC++ [67], which is a library extension to the C++ programming language with state-of-the-art HPC capabilities for parallel computation. UPC++ runs on a variety of platforms, including distributed memory supercomputers and networked clusters. Consequently, the SIMCoV application can run on systems from laptops up to high performance supercomputers without requiring any code changes. Because the computational cost of the simulation is substantial, only small areas can be simulated on smaller systems with one or a few compute nodes. Simulations of significant fractions of the human lung require hundreds or thousands of compute nodes.

## Massively parallel simulation

SIMCoV was designed by composing parallelizable functions to create a highly scalable simulation framework. Running immense agent-based simulations of this type would not be computationally feasible without the application of high performance computing techniques (HPC). For SIMCoV, we relied on the HPC primitives and multi-process communication techniques provided by UPC++. We reduced the communication and computational work load of running large SIMCoV simulations using different approaches. To mitigate expensive cross-process calculation, we maintain two- or three-dimensional blocks of closely spaced grid points in contiguous single process memory. This lowers the chance that for any given update of agents in the simulation that two processes have to share memory. To prevent unnecessary computation costs, we maintain an *active list* of agents and grid points that are updated at each iteration of the simulation. Some grid points can be predicted to remain constant from one iteration to the next, so there is no need to update their behavior for that step.

SIMCoV can be used to run large simulations in minutes on suitable supercomputers. For example, we simulated an infection seeded with multiple initial FOI in a 285000 x 285000 x 5 $\mu m^3$ tissue, comprising over three billion epithelial cells, which is equivalent to a single slice of tissue through the full human lung. On the NERSC Cori supercomputer, it took 15 minutes to complete a two-week simulation on 32 nodes.

## ODE model implementation

Our ODE implementation replicates the model in [12], which was fit to the published sputum swab viral load data of the patients in [36]. Most parameters are similar to our SIMCoV parameters, such as the virion clearance rate and virion production rate. The ODE parameters were fit

to the population of patients using non-linear mixed effects modeling in [12]. To account for the variation in patients, they fit each individual patient, varying cell death rate, infectivity, and virion production rate. The model relies on target cell limitation to fit the viral peak for each patient, and fits were assessed using AIC. The ODE fits in Fig 6 use the parameters selected as best fits for each individual patient. We implemented this ODE model using the scipy odeint Python package, the model description in [12], and the fit model parameters for patient A in [12].

## Acknowledgments

We thank Ruian Ke for sharing insights and an early draft of his ODE model; Matthew Fricke, UNM CARC, and ASU RC for HPC support; and Greg Forest for helpful discussions.

## Author Contributions

**Conceptualization:** Melanie E. Moses, Steven Hofmeyr, Judy L. Cannon, Stephanie Forrest.

**Funding acquisition:** Melanie E. Moses, Steven Hofmeyr, Judy L. Cannon, Stephanie Forrest.

**Investigation:** Melanie E. Moses, Steven Hofmeyr, Judy L. Cannon, Akil Andrews, Rebekah Gridley, Monica Hinga, Kirtus Leyba, Abigail Pribisova, Vanessa Surjadidjaja, Humayra Tasnim, Stephanie Forrest.

**Methodology:** Melanie E. Moses, Steven Hofmeyr, Judy L. Cannon, Akil Andrews, Rebekah Gridley, Monica Hinga, Kirtus Leyba, Abigail Pribisova, Vanessa Surjadidjaja, Humayra Tasnim, Stephanie Forrest.

**Project administration:** Melanie E. Moses, Steven Hofmeyr, Judy L. Cannon, Stephanie Forrest.

**Software:** Steven Hofmeyr, Akil Andrews, Kirtus Leyba, Abigail Pribisova, Vanessa Surjadidjaja, Humayra Tasnim.

**Supervision:** Melanie E. Moses, Steven Hofmeyr, Judy L. Cannon, Stephanie Forrest.

**Validation:** Melanie E. Moses, Steven Hofmeyr, Judy L. Cannon, Akil Andrews, Rebekah Gridley, Monica Hinga, Kirtus Leyba, Abigail Pribisova, Vanessa Surjadidjaja, Humayra Tasnim, Stephanie Forrest.

**Visualization:** Steven Hofmeyr, Akil Andrews, Kirtus Leyba, Abigail Pribisova, Vanessa Surjadidjaja.

**Writing – original draft:** Melanie E. Moses, Steven Hofmeyr, Judy L. Cannon, Akil Andrews, Monica Hinga, Kirtus Leyba, Abigail Pribisova, Humayra Tasnim, Stephanie Forrest.

**Writing – review & editing:** Melanie E. Moses, Steven Hofmeyr, Judy L. Cannon, Abigail Pribisova, Stephanie Forrest.

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
