## [Decision Letter · Decision Letter 0]

19 Aug 2021

Dear Dr. Moses,

Thank you very much for submitting your manuscript "Spatially distributed infection increases viral load in a computational model of SARS-CoV-2 lung infection" for consideration at PLOS Computational Biology.

As with all papers reviewed by the journal, your manuscript was reviewed by members of the editorial board and by several independent reviewers. In light of the reviews (below this email), we would like to invite the resubmission of a significantly-revised version that takes into account the reviewers' comments.

We cannot make any decision about publication until we have seen the revised manuscript and your response to the reviewers' comments. Your revised manuscript is also likely to be sent to reviewers for further evaluation.

Sincerely,

Amber M Smith

Associate Editor

PLOS Computational Biology

Rob De Boer

Deputy Editor

PLOS Computational Biology

Reviewer's Responses to Questions

**Comments to the Authors:**

Reviewer #1: Full review is uploaded as attachment.

Reviewer #2: In this study, the authors present a large-scale agent-based model to simulate and investigate infection and immune dynamics in lung epithelial tissue during SARS-CoV-2 infection. The Spatial Immune Model of Coronavirus (SIMCoV) considers the dynamics of lung epithelial cells, SARS-CoV-2 viral load, inflammation signals and CD8+ T cells within a spatio-temporal context. Analyzing their modeling environment they find that heterogeneity in patient outcomes could be explained by the number of initial fields of infection within lung tissue, as well as the timing of the T cell response.

The article is generally well written and structured. However, some aspects regarding the analyses and methodology are only insufficiently explained and would need some further explanations. This especially relates to the parameterization of the agent-based model, which is a key aspect for such highly complex models and its comparison to patient data. In addition, the analysis seems a bit limited in terms of making use of the stochasticity that an ABM, especially one considering the spatial distribution of cells, can provide.

# Major points:

(1.) SIMCoV represents an extended agent-based modeling environment to simulate and study viral infection within tissue in a spatially defined context. It is claimed that considering infection dynamics in a spatially-resolved context helps to explain the variation observed in individual patient outcome to SARS-CoV-2 infection, with the number of initially infected sites being an important factor determining disease progression. To which extent could this not also be covered by different initial viral loads within an ODE-model, as well as different timing of the T cell responses? While it is investigated how the distribution of the same number of infected cells within different numbers of initial fields of infection influence the dynamics, I could not follow the claim that this analysis might provide a better explanation of variation in patient outcome than the analysis provided by the ODE model. Both seem to explain the data comparably well, judged from the RMSE (see also point 3).

(2.) Agent-based models usually provide stochastic outputs given repeated simulations with the same parameter set. However, in each of the figures (e.g. Figure 3,4,5,7) only a single line for the model simulations of a particular scenario is shown. It is not clear or specified within the figure legends if this is the mean over several simulation runs or a single representation. In any case, it would be very important to know how much variation in the dynamics can occur from one given model parameterization. Especially the distribution of the initial fields of infection will potentially have a substantial impact on the observed dynamics in such a spatial modeling environment. In this regard, the authors should investigate how non-equidistant distribution of FOI (see Fig. 3a) would affect the model outcome and variation. The authors need to report the variation in model outcomes to corroborate their claims, i.e. that the initial number of FOI partly determines disease outcome.

(3.) Figure 5: How was SIMCoV adapted to the patient data? On page 6, bottom, the phrasing leaves the impression that SIMCoV was actually “fitted” to the data. What kind of fitting procedure was used for this task? From the Materials and Methods section, I get more the impression that several parameters were varied and manually adapted according to values and parameter ranges from the literature. But it is not clear which parameters were fixed and which ones were varied/fitted/adapted. If only the initial FOI was varied to explain the patient data in Figure 5 (p.6/7) it is also not clear if FOI were always equidistantly (as shown in Figure 3a) or randomly placed within the grid. If several parameters were varied, are the parameters identifiable or are they correlated given the complexity of the model? If the simulations were manually adapted, the authors might want to consider automatic parameter inference methods, such as approximate Bayesian computation (ABC-methods), as they have been used before for agent-based models to analyze multi-cellular systems and infection dynamics (e.g. Jagiella et al. Cell Systems 2017, Imle et al. Nat Com. 2019). Parameterizations of such modeling environments is one of the key aspects for obtaining reliable predictions and simulations.

(4.) Some details on how the ABM was implemented are missing. While the different processes are explained and introduced, it is not fully specified how these processes are actually parameterized, i.e. how probabilities of events were calculated and simulated. While it is very applaudable that the authors make their simulation environment publicly available, appropriate details on how the parameters were used within the code seem to be missing, if I did not miss this within the text.

(5.) The ODE-model used to explain the individual patient data is not introduced, nor the methods how this was fitted to the data (e.g. mixed-effect model?, individual fits?). Even though this might be published previously, I think the details have to be repeated within the manuscript to compare the different modeling approaches and evaluate the outcomes. In addition, confidence intervals should be given for the estimates within Table 1.

# Minor points:

- page 2, bottom: If epithelial cells are modeled as a 2D grid, it is not clear why they have to be represented as a volume. Does this play a role for e.g. diffusion?

- Figure 7: How comparable are the different scenarios when studied in 2D and 3D? Were the same number of cells/ target cell densities used, as well as the same number of T cells able to infiltrate the tissue?

- Figure 3d: I am not sure how much informative the peak number of T cells would be on viral control in order to compare the different settings. The peak might occur at different time points and the overall level of the T cell responses might be similar. Wouldn’t the cumulative T cell concentration up to a certain time point, and potentially a ratio between T cells and Viral load be more informative for a comparison between the different scenarios, i.e, clearly indicating that the spatial distribution of the number of locations and not the magnitude of initial infection determines disease progression?

- There have been other studies that introduced multi-cellular modeling environments for studying viral infections within epithelial tissues (Sego et al. PLos Comp Biol. 2020). The authors might want to discuss how their approach relates to those. In addition, the authors should discuss why their model is especially a modeling environment for coronaviruses, and not viral infections per se. Only in the last part when studying lung structure, the modeling environment seems to become lung specific.

Reviewer #3: In this paper, Moses et. al. propose a nicely designed mathematical model that includes spatial aspects of infection to recapitulate the SARS-CoV-2 viral load variability among infected individuals. The model is an agent-based one that simulates infection dynamics and CD8 responses in the space of millions of epithelial cells. This approach is relevant as it may explain how more heterogenous virus dynamics could be produced as a function of the spatial parameters, something that ordinarily differential equations are limited to do in this context. Below are my comments:

1. Regarding the peak viral load as a function of the initial FOI, I was wondering if the proportionality is also affected by spatial distribution of the virions. In other words, if the initial FOI and initial virions on them are randomly distributed in the grid—and not just equally dispersed—would the peak viral load still be proportional? What if all the initial FOI is in a specific place in the grid? In a target cell-limited model, or even those that include effector cells, the peak viral load wouldn’t change as a function of the initial viral load. In that sense, it would be helpful to have a more elaborated mechanistic explanation of why this is not the case in the spatial model. Examples of this would be to have spatial simulations of infected cells and CD8 cells per each case illustrating the possible mechanisms. Is the proportionality only due to more infection sites or due to something else like the dispersion of CD8 cells that decrease the chances to attack the infection?

2. Although authors present the effect of two parameters related to the CD8 response with respect to the virus dynamics, it would be also helpful to do a sensitivity analysis on how that compares to the effect of other parameters in Table 2. Which parameters are driving each phase of the virus dynamics? If there are multiple parameters that drive one aspect of the dynamics, which one is more significant? This is relevant in the sense that it would highlight in which aspects the spatial aspects (diffusion rates, initial distribution of variables in the grid, etc) are more significant in the virus dynamics than the non-spatial ones.

3. After having a set of estimated parameters from the model fits, it would be more convincing, in terms of viral load variability, if the model could be simulated multiple times to show that it does allow for more heterogeneity in terms of peak levels, time to peak, time to shoulder phase and time to control.

4. It seems to me the paper is lacking a section (in the Materials and Methods) describing the fitting procedure in detail.

5. Would the model be able to reproduce multiple viral peaks as in figure 5G?

6. The github page did not work when I tried to open it.

**Have the authors made all data and (if applicable) computational code underlying the findings in their manuscript fully available?**

Reviewer #1: **No: **The link to the code repository was broken. "page not found".

Reviewer #2: Yes

Reviewer #3: Yes

PLOS authors have the option to publish the peer review history of their article (what does this mean?). If published, this will include your full peer review and any attached files.

Reviewer #1: No

Reviewer #2: No

Reviewer #3: No
---

## [Decision Letter · Decision Letter 1]

16 Nov 2021

Dear Dr. Moses,

Thank you very much for submitting your manuscript "Spatially distributed infection increases viral load in a computational model of SARS-CoV-2 lung infection" for consideration at PLOS Computational Biology. As with all papers reviewed by the journal, your manuscript was reviewed by members of the editorial board and by several independent reviewers. The reviewers appreciated the attention to an important topic. The reviews lie between major and minor with important points raised by Reviewer 2 that require addressing. 

Sincerely,

Amber M Smith

Associate Editor

PLOS Computational Biology

Rob De Boer

Deputy Editor

PLOS Computational Biology

[LINK]

Reviewer's Responses to Questions

**Comments to the Authors:**

Reviewer #2: In this revised version, the authors have addressed all of my previous comments by performing additional analyses and extending previous explanations on the used methods within the manuscript. This clearly has improved and clarified the manuscript. However, I still would have some comments that I would be happy to see addressed by the authors.

# Major points:

(1.) Figure 3b): It is quite surprising that even given random positioning of individual initial foci, the variation between different simulation runs is basically non-existent. Given the large dependency of the infection dynamics on the initial foci number this is difficult to believe. As indicated in Figure 2, given the chosen parameterization of viral diffusion and infectivity, the infection seems to be largely spreading to directly neighboring cells, i.e. no seeding of new foci by diffusing virions. This means infection spread depends on the combined “surface area” of all foci. However, if foci were seeded randomly, I would expect that at least some foci are initiated closely together, merge early or are initialized close to the boundary of the grid, so that this effectively decreases the “spread surface”, and, thus, should lead to some variation between the simulations. Did the authors also had a look at the corresponding plot for Figure 3b for the number of infected/uninfected epithelial cells? Did this also not reveal any variation in the dynamics? At least for 16 randomly placed, initial foci, I would expect this.

(2.) Although this might be only semantic, I would recommend to speak of “adapting” the model to data rather than “fitting”. In my view, fitting always implies an automatic/algorithmic approach to minimize a difference between model and data. “Manual fitting” could be misleading in this sense. Although it is indicated that some ordered way of adjusting the parameters for each individual patient is used (novel paragraph in M&M), it is still not an elaborated search of the possible parameter space (i.e. fixed parameter ranges for all patients), as has been acknowledged by the authors.

# Minor points:

- Related to point 1 above: I couldn’t find within the text if for the 2D simulations in Figure 2D and 3, periodic or fixed boundaries were used for the simulations.

- p8 bottom: “… controlled by a even a high ….”

- Fig. 7: How many foci were used to start the infection in the different topologies? I would assume this should be a single infected cell to allow comparison.

Reviewer #3: The authors have addressed all my comments successfully.

My only comment is that the results of Figure 8 seemed a little out of place. It was odd to see results of peak viral load in CD8/viral clearance results and the methods section. My suggestion would be to divide Figure 8 in two:

-The first figure would include how parameters affect the viral peak only and include a reference to those results in the second section of the results, "Peak viral load is proportional...". In this way, it would show better how the effect of FOI in the viral peak is more prominent than the other parameters in the same section. The figure related to the viral peak could be part of Figure 3.

-The second figure would include how parameters affect the final percentage of infected cells (as a measure for viral clearance) and could be part of Figure 4. This figure would be referenced in the "Effect of CD8+ T cell response on viral clearance" section.

In this way, Figure 8 is not at the end/methods, and its results would follow the flow of the paper.

**Have the authors made all data and (if applicable) computational code underlying the findings in their manuscript fully available?**

Reviewer #2: Yes

Reviewer #3: Yes

PLOS authors have the option to publish the peer review history of their article (what does this mean?). If published, this will include your full peer review and any attached files.

Reviewer #2: No

Reviewer #3: No

Figure Files:

Data Requirements:

Reproducibility:

References:

---

## [Decision Letter · Decision Letter 2]

9 Dec 2021

Dear Dr. Moses,

We are pleased to inform you that your manuscript 'Spatially distributed infection increases viral load in a computational model of SARS-CoV-2 lung infection' has been provisionally accepted for publication in PLOS Computational Biology.

Best regards,

Amber M Smith

Associate Editor

PLOS Computational Biology

Rob De Boer

Deputy Editor

PLOS Computational Biology

Reviewer's Responses to Questions

**Comments to the Authors:**

Reviewer #2: The authors have clarified all the previous issues. I have no further comments.

**Have the authors made all data and (if applicable) computational code underlying the findings in their manuscript fully available?**

Reviewer #2: None

PLOS authors have the option to publish the peer review history of their article (what does this mean?). If published, this will include your full peer review and any attached files.

Reviewer #2: No

---

## [Editor Report · Acceptance letter]

21 Dec 2021

PCOMPBIOL-D-21-01181R2 

Spatially distributed infection increases viral load in a computational model of SARS-CoV-2 lung infection

Dear Dr Moses,

I am pleased to inform you that your manuscript has been formally accepted for publication in PLOS Computational Biology. Your manuscript is now with our production department and you will be notified of the publication date in due course.

With kind regards,

Livia Horvath
